# The Sterilization of Human Milk by Holder Pasteurization or by High Hydrostatic Pressure Processing Leads to Differential Intestinal Effects in Mice

**DOI:** 10.3390/nu15184043

**Published:** 2023-09-18

**Authors:** Lionel Carneiro, Lucie Marousez, Matthias Van Hul, Léa Chantal Tran, Marie De Lamballerie, Delphine Ley, Patrice D. Cani, Claude Knauf, Jean Lesage

**Affiliations:** 1INSERM U1220, Institut de Recherche en Santé Digestive (IRSD), Université Paul Sabatier, Toulouse III, CHU Purpan, Place du Docteur Baylac, CS 60039, CEDEX 3, 31024 Toulouse, France; lionel.carneiro@inserm.fr (L.C.); claude.knauf@inserm.fr (C.K.); 2NeuroMicrobiota, International Research Program (IRP) INSERM/UCLouvain, 31024 Toulouse, France; matthias.vanhul@uclouvain.be (M.V.H.); patrice.cani@uclouvain.be (P.D.C.); 3Univ. Lille, Inserm, CHU Lille, U1286-INFINITE-Institute for Translational Research in Inflammation, 59000 Lille, France; lucie.marousez@univ-lille.fr (L.M.); delphine.ley@chu-lille.fr (D.L.); 4Metabolism and Nutrition Research Group, Louvain Drug Research Institute (LDRI), UCLouvain (Université catholique de Louvain), 1200 Brussels, Belgium; 5WELBIO Department, WEL Research Institute (WELRI), Avenue Pasteur, 6, 1300 Wavre, Belgium; 6Division of Gastroenterology Hepatology and Nutrition, Department of Paediatrics, Jeanne de Flandre Children’s Hospital, CHU Lille, 59000 Lille, France; lea.tran@chu-lille.fr; 7GEPEA, UMR CNRS 6144, ONIRIS CS 82225, 44322 Nantes, France; marie.de-lamballerie@oniris-nantes.fr; 8Institute of Experimental and Clinical Research (IREC), UCLouvain (Université catholique de Louvain), 1200 Brussels, Belgium

**Keywords:** human milk, high hydrostatic pressure processing, holder pasteurization, mice, gut barrier, microbiota

## Abstract

Background: Human milk banks (HMBs) provide sterilized donor milk (DM) for the feeding of preterm infants. Most HMBs use the standard method of Holder pasteurization (HoP) performed by heating DM at 62.5 °C for 30 min. High hydrostatic pressure (HHP) processing has been proposed as an alternative to HoP. This study aims to evaluate intestinal barrier integrity and microbiota composition in adult mice subjected to a chronic oral administration of HoP- or HHP-DM. Methods: Mice were treated by daily gavages with HoP- or HHP-DM over seven days. Intestinal barrier integrity was assessed through in vivo 4 kDa FITC–dextran permeability assay and mRNA expression of several tight junctions and mucins in ileum and colon. Cecal short chain fatty acids (SCFAs) and microbiota were analyzed. Results: HHP-DM mice displayed decreased intestinal permeability to FITC–dextran and increased ileal mRNA expression levels of two tight junctions (*Ocln* and *Cdh1*) and *Muc2*. In the colon, mRNA expression levels of two tight junctions (*Cdh1* and *Tjp1*) and of two mucins (*Muc2* and *Muc4)* were decreased in HHP-DM mice. Cecal SCFAs and microbiota were not different between groups. Conclusions: HHP processing of DM reinforces intestinal barrier integrity in vivo without affecting gut microbiota and SCFAs production. This study reinforces previous findings showing that DM sterilization through HHP might be beneficial for the intestinal maturation of preterm infants compared with the use of HoP for the treatment of DM.

## 1. Introduction

Human milk is the best food to meet the nutritional requirements of newborns, particularly the most fragile newborns, such as preterm infants. However, when mother’s own milk is not available or is in short supply, donor milk (DM) can be provided by human milk banks (HMBs) as alternative for the feeding of preterm newborns. In this critical clinical situation, the use of DM compared with milk formula is associated with an improvement in feeding tolerance and with a drastic reduction in the incidence of sepsis and necrotizing enterocolitis (NEC), one of the most prevalent and devastating gastrointestinal disorders observed in preterm infants [1]. To ensure the microbiological safety of DM, most HMBs perform a sterilization of DM. Most HMBs use the standard method of Holder pasteurization (HoP), performed by heating milk to 62.5 °C for 30 min in pasteurizers designed for this purpose [2]. This thermal treatment is considered to be a good compromise between microbiological safety and the maintenance of the nutritional quality of DM. However, several studies have recently demonstrated that HoP reduces some nutritional compounds of DM [3,4]. In addition, it has been demonstrated that this pasteurization also degrades numerous heat-sensitive and bioactive factors, such as immunoglobulins, lactoferrin, some vitamins, lysozyme, the bile salt-dependent lipase (BSSL) and several important metabolic hormones [5,6,7]. Therefore, the European Association of Human Milk Banks (EMBA) recommends researching and implementing innovative processing methods for the sterilization of DM in HMBs [8].

Various methods have been described for the sterilization of DM, such as high-temperature short-time pasteurization, ultraviolet irradiation or high hydrostatic pressure (HHP) processing [5]. Over the last 30 years, HHP processing has been used in the food industry to achieve the microbial decontamination of foods [9]. Recent evidence has demonstrated that HHP may be one of the best innovative methods to sterilize DM, as this method maintains the nutritional value of DM as well as numerous bioactive compounds such as immunoglobulins, lactoferrin, lysozyme, BSSL enzyme, milk oligosaccharides and several hormones close to their initial levels (i.e., those found in raw milk) [5,7,10,11]. Recently, we have demonstrated that the use of a moderate HHP protocol (four cycles of 5 min at pressure of 350 MPa, performed at 38 °C) remarkably preserved these important milk compounds [4,7,10,11].

In preterm infants, the intestine is a very vulnerable organ due to its structural immaturity and to the weak integrity of the intestinal barrier (i.e., mucus production, permeability, gut immunity) [1,12,13]. These early alterations can lead to NEC, which affects 1% to 12% of preterm newborns and with a higher risk in newborns whose birth weight is less than 1500 g [12,13]. NEC progresses to an acute development of necrosis in segments of the small and large intestine followed by the development of a severe systemic sepsis [12,13]. Several compounds of human milk, such as oligosaccharides, growth factors, hormones or bioactive lipids, exert protective effects on the intestinal barrier, such as on its permeability, immunity, mucus production and inflammatory state [14]. Thus, the preservation of all milk compounds in sterilized DM is crucial for optimal gut health and for the development of preterm infants. Although numerous biochemical data have shown that different types of sterilization of DM affect the concentration of numerous milk compounds [3,4,5,6,7,10,11]; thus far, no study has evaluated in vivo the consequences of different modes of DM sterilization on the intestinal barrier and microbiota. The present study aims to evaluate in vivo the intestinal consequences of two types of DM sterilization in adult mice subjected to a chronic oral administration of HoP- or HHP-DM over seven days. Intestinal paracellular permeability was measured in vivo after an oral administration of FITC-dextran in a first group of treated mice. In a second group, the expression of several markers of intestinal barrier integrity was quantified using RT-qPCR in the ileum and colon; cecal short chain fatty acids (SCFAs) levels and microbiota were also analyzed.

## 2. Materials and Methods

### 2.1. Milk Collection and HoP and HHP Processing

Frozen DM samples from 8 donors were provided by the regional HMB (Lactarium Régional de Lille, Jeanne de Flandre Children’s Hospital, CHU Lille). After thawing, all milk samples were pooled, and 2 different batches of DM were created. One batch of DM was subjected to HoP according to the standard pasteurization protocol (62.5 °C for 30 min) in our reginal HMB; the second batch was subjected to HHP processing as previously described [10]. The set of HHP parameters was as follows: pressure = 350 MPa, temperature = 38 °C, VA (application rate) = 1 MPa.s^−1^, number of cycles = 4 cycles, duration of each cycle = 5 min. Sterilized DM samples using HoP and HHP processing were stored at −80 °C until used.

### 2.2. Mice, Tissues Collections and Intestinal Permeability Assay

Nine-week-old male C57BL/6J mice (n = 40) (Charles River Laboratory, l’Arbresle, France) were housed individually in a controlled environment (room temperature of 23 °C ± 2 °C, 12:12 h light-dark cycle hours’ day-light cycle) and had ad libitum access to food and water. Mice were divided into 2 groups (n = 20/group) and orally supplemented with HoP- or HHP-DM (100 μL/day) over 7 days. Ten mice per group were sacrificed by cervical dislocation under fed conditions. Ileum and distal colon were collected, washed, flash-frozen and stored at −80 °C for RT-qPCR experiments. Cecal contents were divided into two samples (for SCFAs and microbiota analyses), flash-frozen and stored at −80 °C until analyses. The remaining mice (10 per group) were used to assess in vivo intestinal paracellular permeability. After an overnight fast, mice were gavaged with 4 kDa FITC–dextran (FD4, 440 mg/kg body weight in PBS, Sigma-Aldrich, Saint-Quentin-Fallavier, France). Blood was collected after four hours by cardiac puncture under anesthesia and centrifuged (10,000× *g* for 10 min). Serum FITC–dextran concentration was determined by fluorometry at 485 nm using FLUOstar Omega microplate reader (BMG Labtech, Ortenberg, Germany). Mice were killed by cervical dislocation at the end of cardiac puncture.

### 2.3. Gene Expression

Gene expression analysis was performed as previously described [15]. Briefly, total RNA was extracted using GenElute Mammalian Total RNA Miniprep Kit (Sigma-Aldrich, Saint-Quentin-Fallavier, France), reverse transcribed using a reverse transcriptase kit (Invitrogen, Illkirch, France) and random hexamers (Invitrogen, Illkirch, France) and analyzed with SYBR Green Real-Time PCR Master Mixes (Thermo Fisher Scientific, Illkirch-Graffenstaden, France) using a LightCycler 480 (Life Technologies, Saint-Aubin, France) and specific primers (Table 1). Gene expression was quantified using the comparative Ct (threshold cycle) method, and results were normalized to HPRT expression.

### 2.4. Cecal SCFAs Quantification

Cecal content samples (10 mg) were homogenized using Precellys in 0.5 mL of a 5 mM NaOH solution containing internal standards (Acetate-D3, Propionate-D2, Butyrate-13C2 and Valerate-D9). Super-natant (300 µL) was mixed with 500 µL propanol/pyridine solution (3:2 *v*/*v*). SCFAs were derivatized using isopropylchloroformate and extracted using 0.5 mL of hexane. Quantification of SCFAs was performed with a gas chromatograph coupled to a mass spectrometer (ISQ LT, ThermoFisher Scientific, Waltham, MA, USA) and XCalibur QuanBrowser software 4.2. (ThermoFisher Scientific).

### 2.5. Microbiota Analysis

The intestinal microbiota was analyzed from cecal content samples. Genomic DNA was extracted using a QIAamp DNA Stool Mini Kit (Qiagen, Hilden, Germany). The V4 region of the bacterial 16S rRNA gene was amplified using the primers 515F (GTGYCAGCMGCCGCGGTAA) and 806R (GGACTACNVGGGTWTCTAAT). Purified amplicons were sequenced using a MiSeq following the manufacturer’s guidelines. Sequencing and demultiplexing was performed at MR DNA (Shallowater, TX, USA). Sequences were processed using QIIME2 (version 2023.2) [16]. The pipeline included primer removal and denoising using DADA2 to obtain the amplicon sequence variant (ASV) table [17] and singletons (ASV present < 2 times). Sequences were clustered based on a 0.99% identity and chimeras were removed using the UCHIME algorithm (implemented in QIIME’s vsearch plugin). Taxonomic classification was performed using a pre-trained naive Bayes classifier implemented in QIIME2 against the SILVA 138 reference database (silva138_AB_V4_classifier.qza) [18]. Reads classified as mitochondria and chloroplast were filtered out while unassigned ASVs are retained. Taxa that could not be identified on genus-level are referred to by the highest taxonomic rank identified.

### 2.6. Statistics

Data are expressed as means ± S.E.M (standard error of mean). Statistics were derived using Graphpad Prism software (version 9.00; San Diego, CA, USA). Statistical differences were assessed by unpaired t-test or Mann–Whitney tests according to sample normality. Principal coordinate analysis (PCoA) was analyzed by PERMANOVA test and Kruskal–Wallis test followed by Dunn’s multiple comparison test. Alpha-diversity indexes (Shannon and Faith) and beta diversity index (Bray Curtis) were analyzed using Kruskal–Wallis and PERMANOVA statistical test, respectively. A *p*-value < 0.05 was considered significant.

## 3. Results

### 3.1. Effect of Treatment of Mice with HoP- and HHP-DM on Gene Expression Level of Selected Markers of the Intestinal Barrier in the Ileum

Analyses were performed after seven days of a daily oral gavage of mice with HHP- or HoP-DM. No significant differences of body weight were observed between groups after treatments. In the ileum of HHP-DM treated mice, the gene expression level coding for some cellular tight junctions including Ocln and Cdh1 was significantly increased (Figure 1a,b), though this was not the case with Tjp1 (Figure 1c). While the expression of Cln3 tended to be increased (*p* = 0.0503) (Figure 1d), other claudins, such as Cln4 and Cln7, were not affected (Figure 1e,f). The gene expression level of mucins was increased for Muc2 in the HHP-DM group (Figure 1g) but not for Muc3 and Muc4 (Figure 1h,i).

### 3.2. Effect of Treatment of Mice with HoP- and HHP-DM on Gene Expression Level of Selected Markers of the Intestinal Barrier in the Colon

Conversely to ileum, in the colon of HHP-DM treated mice the gene expression level of Ocln was not altered (*p* = 0.0736) compared with HoP-DM treated mice (Figure 2a). Moreover, while the gene expressions of Cdh1 and Tjp1 were decreased in HHP-DM mice (Figure 2b,c), the gene expression level coding for the claudins Cln3, Cln4 and Cln7 was unaffected (Figure 2d–f). For mucins, Muc2 expression was reduced in HHP-DM mice (Figure 2g), Muc3 was not affected (Figure 2h), and Muc4 expression was reduced compared with HoP-DM mice (Figure 2i).

### 3.3. In Vivo Intestinal Paracellular Permeability in HoP- and HHP-DM Mice

To assess in vivo intestinal paracellular permeability, fasted mice were gavaged with 4 kDa FITC–dextran and serum FITC–dextran concentrations were measured after a 4 h period. HHP-DM treated mice showed a significant decrease in intestinal paracellular permeability compared with HoP-DM mice (Figure 3).

### 3.4. Cecal SCFAs Levels in HoP- and HHP-DM Mice

Six major SCFAs were measured in the cecum of mice (Figure 4). The most concentrated SCFAs (acetate, propionate and butyrate) and the least concentrated SCFAs (isobutyrate, valerate and isovalerate) were not significantly different between experimental groups (Figure 4a–f).

### 3.5. Microbiota Analysis in the Cecum of HoP- and HHP-DM Mice

Principal coordinate analysis (PCoA) was carried out to evaluate the variation of gut microbiomes between the groups (beta diversity) (Figure 5a). The PCoA plot based on the Bray–Curtis dissimilarity distances showed similarity (overlapping clustering) between the microbial composition from the two treatments (PERMANOVA test, *p* = 0.35).

Alpha diversity indexes (Shannon and Faith’s phylogenetic diversity (PD)) were analyzed to investigate the richness and evenness of gut microbiome between treatments. There were no significant differences of bacterial community diversity (Shannon, *p* = 0.45, Kruskal–Wallis test) and phylogenetic distance between OTUs in each group (Faith’s PD index, *p* = 0.36, Kruskal–Wallis test) (Figure 5b). Overall, no major differences were found between the two experimental groups, suggesting that the processing method used for the sterilization of DM (HoP or HHP) did not significantly alter the gut microbiota of treated mice (Figure 5c).

## 4. Discussion

The intestinal barrier is essential in early life to prevent infection, inflammation, and food allergies. This barrier consists of several actors including an epithelial layer, a mucus layer, the immune system and microbiota but also antimicrobial peptides and secretory immunoglobulin A (sIgA) that protect the intestinal mucosa [14]. The weak integrity of the intestinal barrier in preterm newborns leads these infants to a high risk to develop NEC and systemic sepsis [12,13]. This is explained mainly by an underdeveloped intestinal immune system and an increased barrier leakiness in these preterm infants [12,13]. Human milk is the gold standard for the feeding of these vulnerable infants. Indeed, in addition to its nutritional content, breast milk contains many bioactive factors that help to establish an optimal intestinal barrier by acting at different cellular and molecular levels such as intestinal epithelium, immune system, mucus production and gut microbiota establishment [14].

The first aim of our study was to evaluate in vivo the consequences of two types of milk sterilization using HoP or HHP processing on the gene expression of some markers of the intestinal barrier and on intestinal paracellular permeability. We observed that mice treated with HHP-DM presented a reduced intestinal paracellular permeability measured after an oral administration of FITC–dextran. These mice also had an increased expression of some ileal tight junctions (*Ocln*, *Cdh1*) and an augmented expression of the intestinal mucin *Muc2*. Interestingly, in a recent study of our group, we showed that HHP-DM mice also displayed a stimulated antioxidant defense and a reduced expression of inflammatory markers in the ileum, compared with HoP-treated mice [19]. It is now well shown that the intestinal barrier function is strongly regulated by the redox homeostasis [20]. Thus, these findings demonstrate that DM treated by HHP processing may reinforce both the intestinal barrier integrity and antioxidant defenses in the small intestine compared with HoP-DM. These effects could be due to the preservation, through HHP processing, of several bioactive milk compounds involved in these systems, such as γ-tocopherol or some hormones, such as GLP-1, insulin and leptin [7,20]. Surprisingly, in the colon of mice treated with HHP-DM, the expression of two tight junctions (*Cdh1* and *Tjp1*) was reduced as well as the expression of the two mucins (*Muc2* and *Muc4*) in comparison with HoP-DM treated mice. These data suggest that HHP-DM may exert opposite effects in the small and large intestine on the intestinal barrier. One hypothesis to explain opposite variations of some tight junctions and mucin expressions between ileal and colonic segments in HHP-DM mice is that HHP-DM may interact in the colon with some bacterial species and generate microbial metabolites that may ultimately affect some intestinal barrier components [21]. As an example, the microbiota in the cecum and colon of mice produces large amount of SCFAs and tryptophan metabolites that are known to be strong modulators of the mucosal epithelial barrier [14,22,23,24].

The second aim of our study was to evaluate the consequences of two types of sterilization of DM on the gut microbiota and intestinal SCFAs production. So far, no study has yet analyzed the effect of different modes of sterilization of DM on the gut microbiota in rodent models and in humans. We have demonstrated that different modes of sterilization of DM do not induce marked differences in microbiota in terms of diversity and abundance of several bacteria in adult mice. Similarly, we report no significant variations of cecal SCFAs concentrations between groups. These findings are interesting as the early intestinal microbiota is an important factor for gut health during the neonatal period. Indeed, early colonization of the gut is an essential process, as it is from this “first” microbiota that the “adult” microbiota will emerge, develop and be maintained throughout life [25]. However, early gut microbiota composition is influenced by several factors, such as birth mode, feeding practices or antibiotic exposure [26,27]. Among these, breastfeeding is a major determinant of the early gut microbiota composition in the first months of life [28,29]. However, it has been shown that certain commensal intestinal bacteria known to promote the development of the intestinal barrier by enhancing the expression of tight junctions in intestinal epithelial cells are highly present in breast-fed infants (e.g., *Bifidobacterium infantis*, *Bifidobacterium bifidum* and *Lactobacillus rhamnosus*) [30,31]. In addition, the presence of these intestinal bacteria in breastfed infants also helps to prevent pathogenic intestinal colonization by competitively excluding pathogens, stimulating the production of antimicrobial peptides at the mucosal level and inhibiting nuclear factor-κB (NF-κB) signaling to attenuate inflammatory responses [32].

Finally, considering the absence of cecal microbiota and SCFAs variations between our two experimental groups, we cannot conclude here on the imbalance between ileal and colonic intestinal barrier gene-expressions in HHP-DM mice. To test the paracellular intestinal permeability, we used 4 kDa FITC–dextran that was administered orally to mice. However, it is difficult to know whether this analysis reflects small intestine or colonic permeability, as the kinetics of 4 kDa FITC–dextran passage into the blood depends on many factors, including fasting time and mouse strains, making conclusions and comparisons between studies complex [21,33]. Moreover, while the composition of the cecal and fecal microbiota appears to differ only moderately, differences in functional and metabolic activities were found between these cecal and fecal microbiota in rodents [34]. As such, because we have investigated the composition of the cecal microbiota, it is plausible that the discrepancies observed between the colonic segment and ileum could be due to differences in the composition or activities specific to the fecal microbiota. Thus, further studies are needed to clarify this point.

## 5. Conclusions

To conclude, we showed that HHP processing of DM reinforces the intestinal barrier function in vivo without affecting gut microbiota and SCFAs production compared with DM sterilized by HoP. This study reinforces previous findings that show that DM sterilization through HHP might be more appropriate than DM treated by HoP for the feeding of preterm infants.

## Figures and Tables

**Figure 1 nutrients-15-04043-f001:**
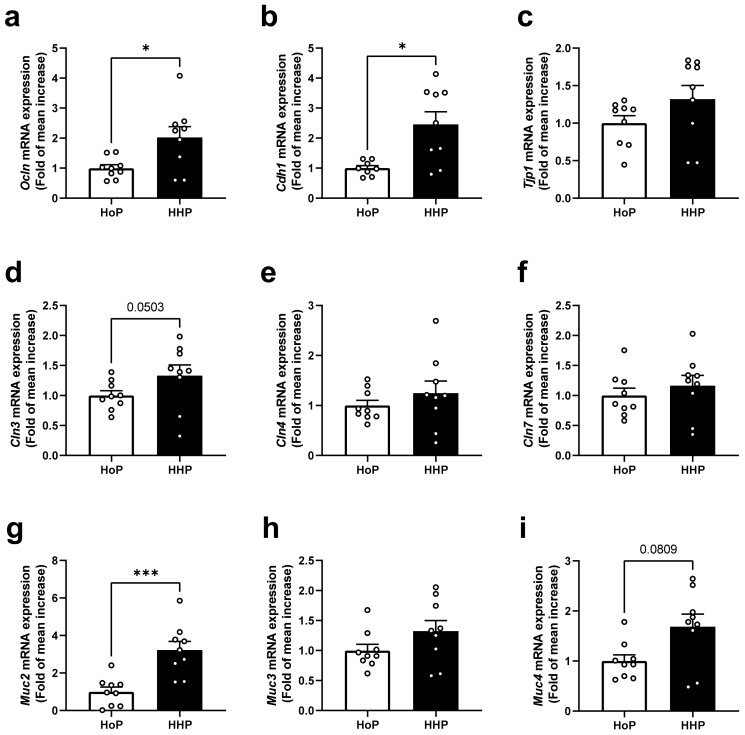
Gene expression levels of markers of barrier integrity in ileum of mice following a 7 day gavage with HoP- or HHP-DM. (**a**) *Ocln* (Occludin), (**b**) *Cdh1* (E-cadherin), (**c**) *Tjp1* (Tight junction protein 1), (**d**) *Cln3* (Claudin 3), (**e**) *Cln4* (Claudin 4), (**f**) *Cln7* (Claudin 7), (**g**) *Muc2* (Mucin 2), (**h**) *Muc3* (Mucin 3), (**i**) *Muc4* (Mucin 4). Circles indicated individual data in the group. n = 10 in each group. * *p* < 0.05, *** *p* < 0.001 HHP vs. HoP group.

**Figure 2 nutrients-15-04043-f002:**
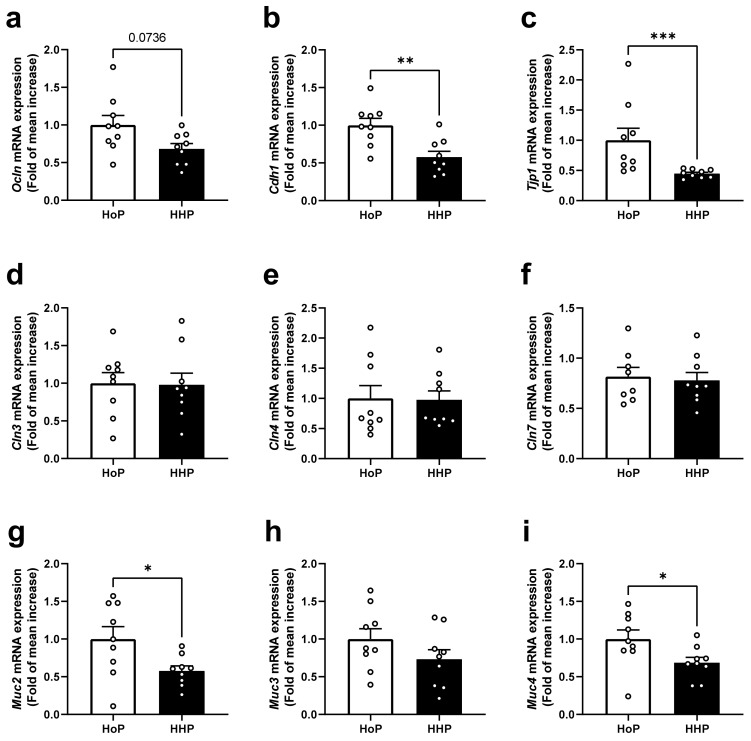
Gene expression levels of markers of barrier integrity in colon of mice following a 7 day gavage with HoP- or HHP-DM. (**a**) *Ocln* (Occludin), (**b**) *Cdh1* (E-cadherin), (**c**) *Tjp1* (Tight junction protein 1), (**d**) *Cln3* (Claudin 3), (**e**) *Cln4* (Claudin 4), (**f**) *Cln7* (Claudin 7), (**g**) *Muc2* (Mucin 2), (**h**) *Muc3* (Mucin 3), (**i**) *Muc4* (Mucin 4). n = 10 in each group. * *p* < 0.05, ** *p* < 0.01, *** *p* < 0.001 HHP vs. HoP group.

**Figure 3 nutrients-15-04043-f003:**
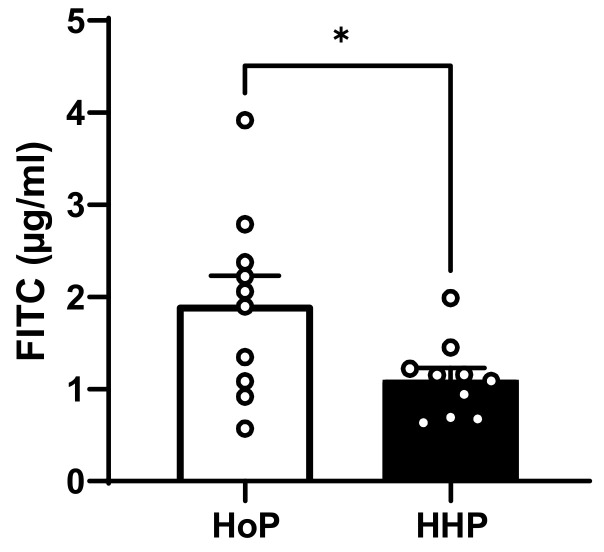
In vivo intestinal paracellular permeability to 4 kDa FITC–dextran in HoP- and HHP-DM mice. n = 10 in each group. * *p* < 0.05 HHP vs. HoP group.

**Figure 4 nutrients-15-04043-f004:**
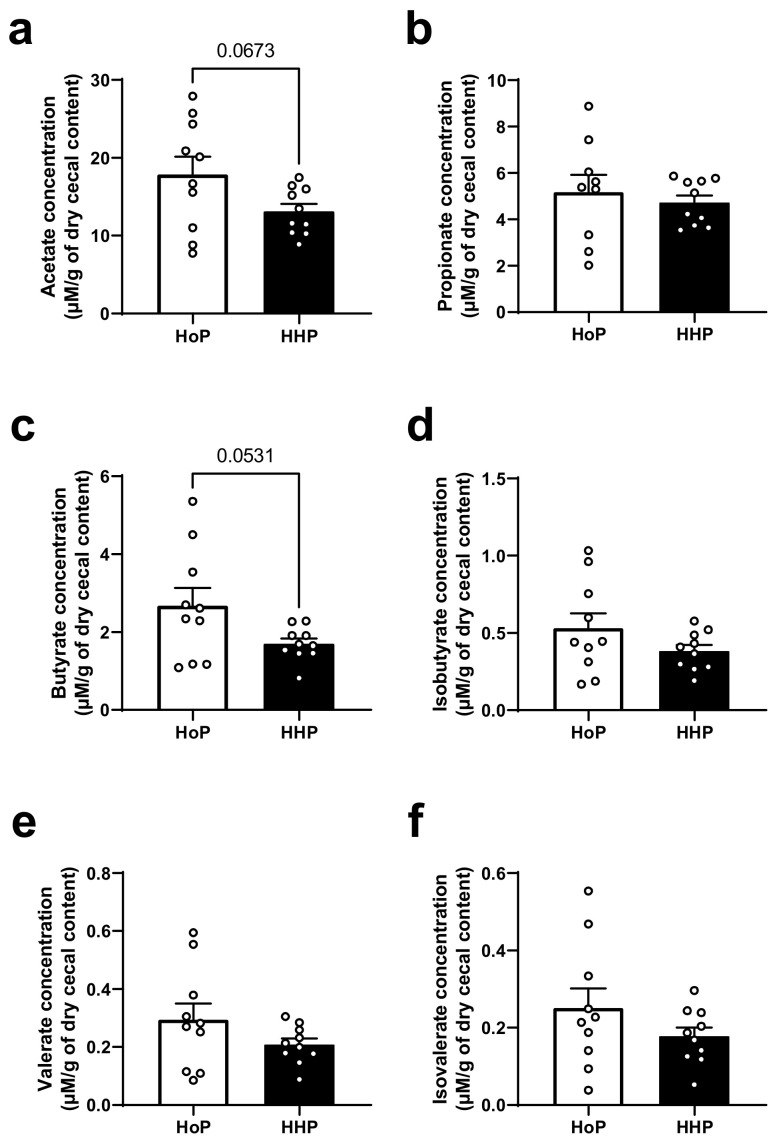
Cecal short chain fatty acid (SCFA) concentrations in mice following a 7 day gavage with HoP- or HHP-DM. (**a**) acetate, (**b**) propionate, (**c**) butyrate, (**d**) isobutyrate, (**e**) valerate, (**f**) isovalerate concentrations are expressed as micromolar (µM) per gram (g) of dry cecal content. n = 10 in each group.

**Figure 5 nutrients-15-04043-f005:**
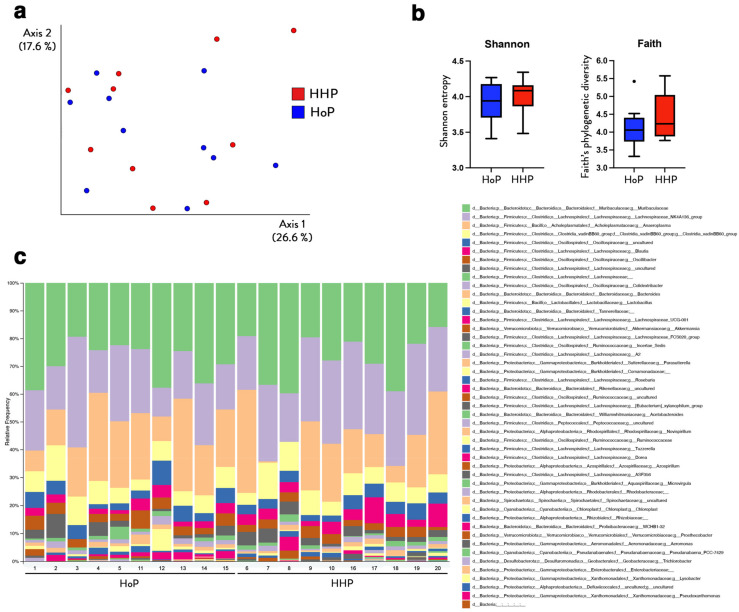
Cecal gut microbiota composition in mice following a 7 day gavage with HoP- or HHP-DM. (**a**) Principal coordinate analysis (PCoA) plot representing beta diversity as Bray–Curtis distances for the gut microbiota of individual mice at the end of the treatment. Samples from HoP and HHP are labeled with blue and red, respectively. (**b**) Boxplots showing distribution of Shannon and Faith’s phylogenetic alpha diversity measures in HHP and HoP samples. (**c**) Stacked bar plot showing taxonomic relative abundance distribution of cecal microbial communities at the genus level with samples sub-categorized by treatment. n = 10 in each group.

**Table 1 nutrients-15-04043-t001:** Primers sequences.

Targeted Gene	Forward Primer	Reverse Primer
*Ocln*	atgtccggccgatgctctc	tttggctgctcttgggtctgtat
*Cdh1*	ccaatcctgatgaaattggaaact	aacaccaacagagagtcgtaag
*Cln3*	ccaggagaggagccgttaag	cccttcgaaaactgacggac
*Cln4*	cgttactccagcgctactc	tcactcagcacaccatgact
*Cln7*	tgatgagctgcaaaatgtacg	ccagggacaccaccattaag
*Tjp1*	gttggtacggtgccctgaaaga	gctgacaggtaggacagacgat
*Muc2*	cggaactccagaaagaagcca	ggcagtcagacgcaaagttgta
*Muc3*	caccttccagccttccctaa	caacgatgtcatgactacctgg
*Muc4*	agaggcagaagaggagtggaga	ggtggtagcctttgtagccatc

## Data Availability

Data of this study are available upon request from the authors.

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
