# Peer review of "The Sterilization of Human Milk by Holder Pasteurization or by High Hydrostatic Pressure Processing Leads to Differential Intestinal Effects in Mice"

_nutrients, 2023, doi:10.3390/nu15184043_

Round 1

Reviewer 1 Report

In a manuscript submitted for review, the Authors reported that sterilization of human milk by holder pasteurization or by high hydrostatic pressure processing leads to differential intestinal effects in mice. I find the subject of the manuscript interesting, the whole work is well thought out. The Authors put a lot of effort into preparing this work.
My comments:
1. the Authors used only males, why not females?
2. line 113: "Twenty mice (n=10/group)...." I find this notation misleading, it should be n=20
3. how were the animals killed?
4. the aim of the work has no clearly formulated research hypothesis.
5. English needs correction

English requires correction, the manuscript contains many grammatical errors

Author Response

R1: In a manuscript submitted for review, the Authors reported that sterilization of human milk by holder pasteurization or by high hydrostatic pressure processing leads to differential intestinal effects in mice. I find the subject of the manuscript interesting, the whole work is well thought out. The Authors put a lot of effort into preparing this work.My comments:

  1. the Authors used only males, why not females?

This study is a first study to evaluate in vivo if two types of sterilization of human milk have differential intestinal effects in mice. Indeed, as preterm newborns are frequently fed with sterilized donor milk, it is important to find the best method of sterilization to preserve all milk compounds for the optimal development and gut health of these vulnerable infants. This question has never been addressed in clinical studies before. In two previous recent studies from our group, we administered HHP-DM to adult male mice. We demonstrated that HHP-DM administration induced a stimulation of antioxidant defenses and reduced some inflammatory markers in both the ileum and liver (Wemelle et al. Antioxidants (Basel) ; 2022; 31; 11(6):1091. doi: 10.3390/antiox11061091); in the second study, we showed that HPP-DM preserves both apelin and GLP-1 and improves glucose tolerance by acting on gut contractions (Wemelle et al. Nutrients ; 2022; 5;14(1):219. doi: 10.3390/nu14010219). Thus, to complete these observations, in the present study, we used (once more) adult male mice to analyze the intestinal permeability and gut microbiota. All these data in adult male males now lead us to pursue new experimental studies but, this time, in developing male and female mice. Currently, some studies are in progress in our laboratory (daily oral administration of HoP-DM or HHP-DM throughout the period of lactation). Some encouraging data show that HHP-DM improves significantly the postnatal growth of growth-restricted mice of both sexes (from PND8-21). Further molecular analyses are also in progress to explain these effects in animals of both sexes.

  1. line 113: "Twenty mice (n=10/group)...." I find this notation misleading, it should be n=20

We have corrected this error in the M&M section.

  1. how were the animals killed?

Animals were killed by cervical dislocation; this was added in the M&M section.

  1. the aim of the work has no clearly formulated research hypothesis.

The aim of the study was more clearly formulated at the end of the introduction (line 84-90).

  1. English needs correction

We have improved the writing of our article with the help of an English-speaking colleague.

Reviewer 2 Report

This manuscript titled “The Sterilization of Human Milk by Holder Pasteurization or by High Hydrostatic Pressure Processing leads to differential intestinal effects in mice”. The comments for this manuscript are as follows:

1.      This manuscript of human milk banks is indeed of great significance. But the main question in this manuscript is how to convince readers to feed mice with human breast milk, and what is the difference in the period? There are differences between humans and mice. Have the authors conducted a blank experiment, such as the difference between mice milk treated with HoP- or HHP-DM (similar to the blank or control experiment). After all, the digestive system of a human is quite different from that of a mouse. For example, if people eat rotten food, most of them will get gastrointestinal problems. However, has anyone ever seen rats get sick from eating rotten food and stop eating it? Probably not. If the authors do not provide a blank or control experiment data, the significance of these data would be greatly reduced (Figure 1- Figure 4).

2.      In addition, what does the meaning of Figure 5 want to prove? I wish that the authors to explain in detail, otherwise the reviewer thinks it is of little significance and can be removed.

3.      In the current technological environment, only measuring mRNA is not enough to prove that it can be translated into protein. Only protein can have a real effect. I don't know why the authors don't measure the expression of protein?

4.      There are many errors in the section of "Refererences". The writing of references should be consistent, and each word should not be capitalized. Please according to the "Instructions for Authors" to rewrite the references.

I decided it should be a major revision.

Author Response

R2:  1.      This manuscript of human milk banks is indeed of great significance. But the main question in this manuscript is how to convince readers to feed mice with human breast milk, and what is the difference in the period? There are differences between humans and mice. Have the authors conducted a blank experiment, such as the difference between mice milk treated with HoP- or HHP-DM (similar to the blank or control experiment). After all, the digestive system of a human is quite different from that of a mouse. For example, if people eat rotten food, most of them will get gastrointestinal problems. However, has anyone ever seen rats get sick from eating rotten food and stop eating it? Probably not. If the authors do not provide a blank or control experiment data, the significance of these data would be greatly reduced (Figure 1- Figure 4).

Of course, the aim of our study was not to test the physiological consequences of feeding mice with human milk but to analyze if two types of sterilization of human milk induce differential intestinal effects in vivo. So far, this question has never been addressed in clinical studies before (due to ethical evidences), thus experimental models have to be used to explore this hypothesis, that’s why we administered DM to adult male mice. In two previous recent studies from our group, we similarly administered HHP-DM to adult male mice. We demonstrated that HHP-DM administration induced a stimulation of antioxidant defenses and reduced some inflammatory markers in both the ileum and liver (Wemelle et al. Antioxidants (Basel) ; 2022; 31; 11(6):1091. doi: 10.3390/antiox11061091); in the second study, we showed that HPP-DM preserves both apelin and GLP-1 and improves glucose tolerance by acting on gut contractions (Wemelle et al. Nutrients ; 2022; 5;14(1):219. doi: 10.3390/nu14010219). Thus, to complete these observations, the present study was focused on the intestinal permeability and gut microbiota. In accordance with your comment, the use of mouse milk treated by HoP or HHP could have been more appropriate but: 1/mouse milk is very different of human in term of composition and probably to sensitivity to HoP and HHP processing, 2/ it is very difficult to obtain large quantities of mouse milk to perform this study. Finally, as our goal was to test the differential intestinal effects of treated-DM, we don’t need other control experiment/group in our study.

  1. In addition, what does the meaning of Figure 5 want to prove? I wish that the authors to explain in detail, otherwise the reviewer thinks it is of little significance and can be removed.

In pediatric units, it is very important to control the establishment of a beneficial gut microbiota especially in preterm infants who are very vulnerable to NEC and sepsis (as explained in the introduction and discussion). We believe that our data on gut microbiota are particularly important for clinicians and for demonstrating in vivo all the digestive effects of two our types of milk sterilizations.

  1. In the current technological environment, only measuring mRNA is not enough to prove that it can be translated into protein. Only protein can have a real effect. I don't know why the authors don't measure the expression of protein?

We agree with this comment however, for the majority of markers studied in Fig 1 and 2, it is very difficult to perform others quantitative analyses. As example, the mucus is composed of 95% of water, several proteins and numerous oligosaccharides. It is also impossible to do histological coloration of it thus the majority of studies only measure mucins mRNAs to evaluate its production. For tight junctions, their protein expression is very low and it is difficult to perform quantitative western blots for numerous of them (histological labelling is also not very appropriate for quantifications). Thus, in our study, we preferred to extract RNA from tissues for RT-qPCR studies to rapidly check numerous actors of the gut barrier. Thus, these results prompted us to test the gut permeability with dex-FITC (in vivo) that is a well-established protocol to test the intestinal barrier permeability.

  1. There are many errors in the section of "Refererences". The writing of references should be consistent, and each word should not be capitalized. Please according to the "Instructions for Authors" to rewrite the references.

References were added with Zotero in accordance with the ACS style guide indicated in the Nutrients web site. We used the same method in our very recent published article (Marousez et al, Nutrients. 2023 Jun 16;15(12):2771. doi: 10.3390/nu15122771). When the accepted article is processed by the journal, the form of the references is indeed different in the final version online. Thus, we did not modify references in this revised version.

Round 2

Reviewer 2 Report

The authors did not correct the reviewer's problems, even the simplest format (References) it still has a lot of errors, I think the current state of this manuscript is not suitable for proceed in Nutrients.

Author Response

comments ref2: "The authors did not correct the reviewer's problems, even the simplest format (References) it still has a lot of errors, I think the current state of this manuscript is not suitable for proceed in Nutrients"

author's reply: We reply to all the comments in the point-by-point letter of our revised article (in the round 1) as well as the form of the references that was in accordance to the recommendations on the web site of the journal.